# The Category of Conventional Physiotherapy: The Case of Parkinson’s Disease Guidelines

**DOI:** 10.3390/jpm12050730

**Published:** 2022-04-30

**Authors:** Martina Hoskovcová, Evžen Růžička, Ota Gál

**Affiliations:** Department of Neurology, Centre of Clinical Neuroscience, First Faculty of Medicine, Charles University, General University Hospital in Prague, 128 21 Prague, Czech Republic; martina.hoskovcova@vfn.cz (M.H.); eruzi@lf1.cuni.cz (E.R.)

**Keywords:** conventional physiotherapy, Parkinson’s disease, guideline, meta-analyses, reviews

## Abstract

This opinion paper reviews the use of the category of “conventional physiotherapy” (CPT) in Parkinson’s disease (PD)-relevant reviews and meta-analyses and points out serious inconsistencies within and among them. These are first discussed in general, leading to the conclusion that, in most cases, the category of CPT encompasses a range of incompatible interventions. This undermines previous conclusions about their superiority or inferiority relative to various other treatment modalities. Next, the update to the European Physiotherapy Guidelines is discussed in detail, since it treats CPT as a global and time-independent category per se, ascribing effects in various domains to it. This introduces several important biases into the findings presented in this publication. These are subsequently discussed, and it is concluded that the categorisation of various physiotherapy techniques under the umbrella term of CPT is empty, or even dangerous, and should be abandoned. Other categories are suggested as a replacement, including “Other Physiotherapy Techniques” and “Multimodal Training”.

## 1. Introduction

Over the last decades, the number of studies investigating physiotherapy (PT) treatment modalities in Parkinson’s disease (PD) has increased exponentially. In parallel, the number of published reviews (Rs) and meta-analyses (MAs) has risen significantly. As is usual for these types of papers, the category of conventional and/or usual physiotherapy/care/exercise—which is subsumed here under the umbrella term of “conventional physiotherapy” (CPT)—is used, typically as one subtype of intervention delivered to the control group, with the other subtypes being sham or no intervention. Consequently, we are presented with conclusions to the effect that a given technique is superior or inferior to CPT. However, while the notion of CPT seems *prima facie* intelligible, it is rarely defined by the authors of the Rs and MAs. Thus, one might wonder what the conclusion about the superiority or inferiority of a given PT modality actually means. Superior or inferior to what exactly? Moreover, when we look at particular Rs and MAs, we often get the feeling that the authors categorise various control group interventions under CPT simply because the authors of the reviewed studies used this classification. This does not, however, necessarily mean that the CPT found in one study is equivalent to—or even vaguely corresponds with—the CPT used in another study. Thus, it is not clear whether it is methodologically sound to create forest plots on this basis or whether the conclusions of such MAs are, in fact, relevant to clinical practice.

In this opinion paper, we examine the use of the category of CPT in recent Rs and MAs for PT interventions in PD, in order to try to identify its definition, if any, and to highlight the potential problems it causes. As an example of a particularly problematic use of CPT, we have chosen the MA by Radder et al. [1], which is discussed in more detail in an attempt to rectify what appear to be erroneous conclusions.

## 2. Review of the Use of the Category of Conventional Physiotherapy in Reviews and Meta-Analyses of Physiotherapy Interventions in Parkinson’s Disease

We have conducted a review of the use of the category of CPT in Rs and MAs of PT in PD using the following search criteria in Pubmed: ((“Parkinson Disease”[Mesh]) AND (“Rehabilitation”[Mesh] OR “Physical Therapy Modalities”[Mesh])). The scope of our search was further limited to reviews, systematic reviews and meta-analyses written in English in the last 5 y (2017–2022). The search identified 220 papers. We excluded 193 papers because they did not contribute anything to the reviewed topic in any meaningful way or were completely irrelevant, leaving 27 papers that were reviewed (Figure 1).

As noted in the “Introduction”, most studies utilise the category of CPT to refer to one type of intervention in the control group, the other being sham or no intervention, e.g., in Flynn et al. [2] or Perry et al. [3]. The description of the intervention in the control group is copied in an uncritical manner from reviewed studies and pasted into the Rs and Mas, which results in some interventions being called “conventional physiotherapy”, others “usual therapy” and yet others “traditional therapy”, even within the same R or MA [4,5,6,7,8,9]. However, there seems to be no difference between these terms, since “conventional” generally means the same thing as “usual” or “traditional”. Moreover, the publications in question do not make any distinction.

In a few studies, some form of definition is suggested or at least a list of techniques is given. For example, Carapellotti et al. [10] and Pinto et al. [11] consider “usual care” to equate with no intervention or medication alone. However, according to Winser et al. [12], “usual care” is an active treatment control, which implies some form of physical activity in the context of a review of the efficacy of a particular PT modality (in this case, Tai Chi). According to Perry et al. [3], “usual care” refers to medication and other programmes without a focus on activities of daily living (ADL) training. By contrast, Cugusi et al. [13] and Luna et al. [14] (with reference to Canning et al. [15]) consider it medical therapy *and* normal activities of daily living or advice to maintain current levels of physical activity respectively.

So far, the meaning of “usual therapy” ranges from no intervention at all to medication and maintaining physical activity levels. However, in the review of cognitive training interventions for dementia and mild cognitive impairment in PD, Orgeta et al. [16] use “usual care” to refer to the treatment that would usually be provided to people with cognitive impairment in PD in the setting in which the study was conducted (including medication, day care, and support, but no specific structured cognitive training intervention). Thus, if we adapt it to PT, CPT could even comprise non-specific and non-structured physiotherapy.

However, Santos et al. [17] consider traditional physiotherapy conventional exercises that encourage the strengthening and/or stretching for the main muscle groups of the body, as well as exercises for cardiovascular fitness. In addition, functional electrical stimulation is considered traditional physiotherapy in this review, which focuses on the use of the Wii video-game console. Therefore, structured and specific PT might also be considered CPT. Similarly, Cugusi et al. [13] define “CPT” as aerobic and strength training programmes, this time in the context of a review of aquatic therapy. Moreover, they define “less conventional PT” as dance, Tai Chi, Nordic Walking, and other complementary therapies and “non-conventional PT” as aquatic exercise alone. In the same vein, Lorenzo-García et al. [18] consider strength, endurance, balance, and gait workouts as CPT in the context of body-weight-supported treadmill training (BWSTT).

These three Rs thus touch upon the essentially relative character of the term CPT. In relation to certain treatment modalities (e.g., aquatic therapy), CPT means something else (e.g., aerobic and strength training programs [13], or any land-based therapy [19,20]) than it does in relation to other interventions like BWSTT (where CPT equals strength, endurance, balance, and gait workouts [18]). In both cases, the listed treatment modalities are specific and structured, but differ depending on the context (see also a different definition of CPT in the context of virtual reality in Lina et al. [21]). The most extreme case is the review of exergaming in PD by Papamichael et al. [22], who consider CPT to comprise joint mobilisation, respiratory balance and coordination exercise, gait or gait-at-home, muscle strength and aerobic exercise, stretching, continuation of ADL, proprioceptive neuromuscular facilitation exercise, general exercises, object manipulation exercises, fall-prevention education programmes, trunk rotation, and transition of the central body—that is, nearly everything. Moreover, the relative character of CPT is determined not only in relation to the category with which it is being compared, but also the geographical context. For example, in the context of Chinese exercise, Qigong might be considered CPT [23].

As can be seen, the definition of CPT varies dramatically (Table 1). This might not necessarily cause any problem in individual studies, which simply show the favourable or unfavourable effect of a given procedure in the experimental group as compared to CPT. Despite the fact that CPT is not usually defined in such studies, it is possible to determine what it was in this particular country at that particular time, at least, theoretically. In our opinion, the problem arises in Rs and MAs of such studies, in which interventions from various different time points and settings are blended together. This becomes even more problematic in MAs when forest plots are created, and a treatment modality is claimed to be superior or inferior to an agglomeration of incompatible or not otherwise specified therapies, like in Alwardat et al. (CPT and Treadmill [7]) or Mackay et al. (unspecified usual care and no therapy [24]).

The worst-case scenario involves utilising CPT as a treatment category per se, as if something like it existed all over the world in a time-independent manner. In fact, Alwardat et al. [7] claim to have identified studies utilising CPT by using “conventional therapies” as a MESH term. However, no such MESH term exists.

## 3. Critical Evaluation of an Example for Implementing an Update to the European Physiotherapy Guidelines

One example that is worth taking a deeper look at is the long-awaited update to the European Physiotherapy Guideline for Parkinson’s Disease (EPGPD) [25], which has recently been published in the form of a meta-analysis by Radder et al. [1].

This MA utilises CPT as one of the classification categories. The motivation for choosing this problematic category was probably the fact that it was used in the EPGPD. However, already in the EPGPD, it had a flawed definition: “All physiotherapist-supervised active exercise interventions targeting gait, balance, transfers or physical capacity, or a combination thereof.” The problem with this definition is that this type of CPT comprises all physiotherapy interventions, because there is no specific difference. Thus, Radder et al. redefine it as “all active (exercise) interventions traditionally used by physiotherapists to manage people with PD, such as traditional physiotherapy techniques or multifaceted interventions combining different physiotherapy techniques.” Here, two specific differences are mentioned: (I) traditional use; and (II) multimodality. However, a counterexample to the former could be cited, namely hand-clapping as a form of auditory cueing, which is the most traditional technique but belongs to the category of “Strategy Training” in the meta-analysis. The latter specific difference, i.e., multimodality, is also not helpful, since, in clinical practice, multimodal training is used all the time and there is no clear link between multimodality and conventional use.

As noted, a crucial problem with using CPT as a classification category is the fact that it is country- and time-specific. In many countries, neurodevelopmental treatment (Bobath) is traditionally used but Radder et al. probably do not want to imply that the results of the CPT category can be applied to this technique. In fact, the whole category of CPT rather gives the impression of being a category of left-over techniques that the authors had trouble classifying. After all, transcranial direct current stimulation in combination with physiotherapy [26] is included in CPT. Moreover, there is no need for such a left-over category, since the authors use the category of “Other Physiotherapy Techniques” to refer to various techniques.

To some extent, using CPT made sense in 2014 (the publication date of the EPGPD) since the point was to contrast modern techniques with the care used at the time in order to optimise it. This is the mission of ParkinsonNet, which the authors of the meta-analysis advocate. Nowadays, however, the situation is different, owing to the enormous awareness-raising campaign that the Dutch team is undertaking worldwide.

Thus, the conclusions of Radder et al. [1] about CPT should be interpreted with caution, because they relate entirely to the studies analysed and not to conventionally used techniques. The impression that CPT improves Movement Disorder Society-Unified Parkinson’s Disease Rating Scale (MDS-UPDRS) III, cadence, 10-m walking test (10MWT), and quality of life (QoL) is very unfortunate because the goal of ParkinsonNet is to endorse innovation and not to preserve the status quo.

Moreover, if you look at the forest plots, then the effect on MDS-UPDRS III is based mainly on studies utilising Schultz autogenic training, aerobic-resistance training, trunk-endurance, and stability training in combination with gait training, and, to some extent, a few other techniques. However, these were either not traditionally used (such as the first one) or are simply combinations of other categories. Similarly, the effect on 10MWT is largely derived from a study utilising Lee Silverman Voice Treatment BIG, which is not part of CPT in many countries. The effect on cadence is based on two studies using smart-bike and an exercise app, where the first is rather aerobic training and the second is, once again, not CPT. Finally, the effect on QoL is again largely dependent on a study utilising lower-limb resistance training and balance training, which is simply a combination of other categories.

In summary, therapies that are behind the alleged effect of CPT are basically multimodal types of training, combining modalities that belong to different categories. Thus, we wonder what the results of the meta-analysis would look like if these studies were categorised differently. However, such a change would also have an impact on other categories. Thus, our view is that the overall results of the paper are undermined.

## 4. Conclusions

For this reason, we recommend abandoning the category of CPT for future Rs and MAs, on the grounds that it is obsolete and misleading. Indeed, the recent American Physical Therapy Association Guideline does not use it [27]. CPT should be replaced by the following categories, depending on what it means in each particular study: (I) If CPT equals a single defined treatment modality (e.g., aerobic training), then it should be classified in this category, despite the fact that the authors of the study in question refer to it as CPT; (II) If CPT is related to a single defined treatment modality which does not have its specific category, e.g., due to its uniqueness (like direct-current stimulation), it can be classified under “Other Physiotherapy Techniques”; (III) If CPT comprises several treatment modalities, it should be classified as “Multimodal Training”.

## Figures and Tables

**Figure 1 jpm-12-00730-f001:**
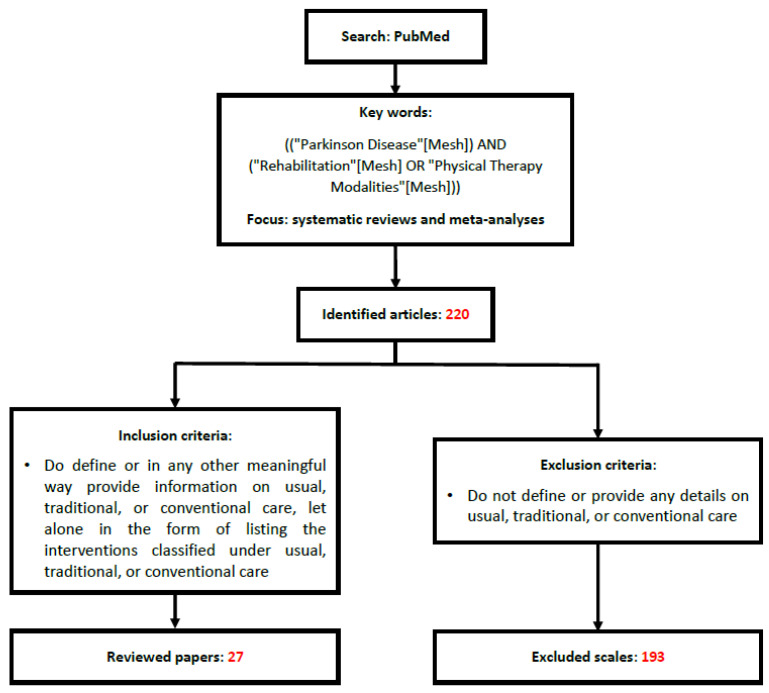
Flowchart of the review process.

**Table 1 jpm-12-00730-t001:** Definition of usual, traditional, or conventional care in the reviewed papers.

			Carapellotti 2020	Pinto 2019	Winser 2018	Perry 2019	Luna 2020	Derived from Orgeta 2020	Santos 2019	Cugusi 2019	Lorenzo-García 2021	Papamichael 2021
Passive treatment control	No intervention		x									
	Medication			x		x	x	x				
Active treatment control	Unspecified active treatment				x							
	Programs without focus on activities of daily living					x						
	Usual activities of daily living						x					x
	Advice						x					
	Non-specific and non-structured physiotherapy							x				
	Structured and specific physiotherapy	Functional electrical stimulation							x			
		Strengthening							x	x	x	x
		Stretching							x			x
		Cardiovascular fitness exercise							x	x	x	x
		Gait training									x	x
		Balance training									x	
		Joint mobilisation										x
		Respiratory exercise										x
		Proprioceptive neuromuscular facilitation										x
		General exercise										x
		Object manipulation exercise										x
		Fall prevention education										x
		Trunk rotation and transition exercise										x

## Data Availability

Not applicable.

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
