# Peer review of "The Category of Conventional Physiotherapy: The Case of Parkinson’s Disease Guidelines"

_jpm, 2022, doi:10.3390/jpm12050730_

Round 1
Reviewer 1 Report
This is a very interesting opinion paper regarding de use of the term of CPT “conventional physiotherapy” in Parkinson’s disease treatment.
The authors rise a very practical problem that CPT had different meaning for different research teams, according to the country and to the time of publishing these date.
In individual papers, the CPT is usually defined by the authors, according to their knowledge and understanding.
A real problem arises when individual studies with different definitions are pooled in reviews and meta-analyses. Moreover, that rises bias in forest plots of meta-analyses when authors rise conclusions about superiority or inferiority of various modalities of PD treatment.
Conclusions drawn are coherent, the term CPT should be replaced as follows:
- a single define treatment modality that should be named (like aerobic training)
- “other physiotherapy techniques” – if there is not a specific category (like direct-current stimulation)
- “multimodal training” – if CPT comprises several modalities
It is a clear, comprehensive paper, with a big relevance in the field.
Author Response
Thank you
Reviewer 2 Report
Physiotherapy is a widely studied treatment paradigm with proven beneficial outcomes for patients. When evaluating physiotherapy interventions in health care studies, the treatment of control groups is often summarized under the term “conventional” physiotherapy. However, there is no common definition for this term and thus, the interventions in the control groups often variy widely between studies. This becomes a problem when comparing various studies using different interventions as baseline groups in reviews and meta-analyses. In the present opinion article, the authors address this problem specifically in studies related to Parkinson’s disease (PD). Given that the existence of a standardized control group is key to a proper interpretation of data and the reproducibility of results, this topic is very relevant also beyond the PD field, especially because it might have a significant impact on treatment recommendations, which directly affect patients.
Comments
The topic fits well into the scope of the Journal of Personalized Medicine and addresses an important definition problem relevant to all scientists and physicians involved in the analysis of physiotherapy strategies. Overall, the problem is clearly stated, and the authors justify their opinion and critique with several current literature examples. However, there are also some minor points of concern. Please find below my detailed comments and suggestions:
- To illustrate the literature search strategy used to find appropriate reviews and meta-analyses, I would recommend showing a flow chart including the number of excluded publications and reason for exclusion.
- Several abbreviations used in the manuscript are not introduced properly. Please indicate the full term before first use of abbreviations (for example: line 70: ADL, line 158: 10MWT)
- To visualize the different definitions of CPT in the examined publications, it would be beneficial to compare them using a table.
- I would suggest matching the title of the 3rd section to the title of the 2nd section by rephrasing, for example: “Critical evaluation of an example for implementing an update to the European Physiotherapy Guidelines”.
- To tailor the key words more specifically to the manuscript, the terms “meta-analyses” and “reviews” could be included.
- The manuscript would benefit from minor spell checks (for example: line 88: TaiChil), and language editing (for example: line 72: insert comma before “respectively”, line 170: “causing” instead of “that are behind”, line 172: “how” instead of “what”).
Author Response
Thank you for your comments. Below find our detailed responses.
Comment 1: To illustrate the literature search strategy used to find appropriate reviews and meta-analyses, I would recommend showing a flow chart including the number of excluded publications and reason for exclusion
Response: We have now included a flowchart. Thank you.
Comment 2: Several abbreviations used in the manuscript are not introduced properly. Please indicate the full term before first use of abbreviations (for example: line 70: ADL, line 158: 10MWT)
Response: Corrected, thank you.
Comment 3: To visualize the different definitions of CPT in the examined publications, it would be beneficial to compare them using a table.
Response: We have now included a table. Thank you.
Comment 4: I would suggest matching the title of the 3rd section to the title of the 2nd section by rephrasing, for example: “Critical evaluation of an example for implementing an update to the European Physiotherapy Guidelines”.
Response: Corrected as suggested. Thank you.
Comment 5: To tailor the key words more specifically to the manuscript, the terms “meta-analyses” and “reviews” could be included.
Response: Both terms were added as key words. Thank you.
Comment 6: The manuscript would benefit from minor spell checks (for example: line 88: TaiChil), and language editing (for example: line 72: insert comma before “respectively”, line 170: “causing” instead of “that are behind”, line 172: “how” instead of “what”).
Response: TaiChi was a typo and we corrected it. Thank you for noticing it. However, regarding the rest of the proposed changes, we are forced to disagree. Our text was proofread by a professional native and his response to the grammar concerns raised is the following:
- line 172: “how X would look” and “what X would look like” are perfectly synonymous expressions. Both are completely standard and they can be used interchangeably. Moreover, if you just replace “what” with “how”, as the reviewer seems to be suggesting, then the result would be “how X would look like”, which is wrong.
- line 170: Merriam-Webster’s definition of “to lie behind” is “to be the cause of (something)” (i.e. it is a straightforward synonym for “causing”). Some other dictionaries add the caveat that it is especially (though not exclusively) used of hidden causes.
- line 72: The “rules” in general for comma use are extremely flexible and vary a lot from writer to writer. Fowler’s, the most authoritative usage guide in the UK (which we are using) does not explicitly address the issue of commas, but if you look at the excerpt below, you will see that not a single example given has a comma around it: “Respectively is correctly used when the sense required is 'each separately or in turn, and in the order mentioned' (e.g. Such a scaffold is fully determined by the parameters R and c specifying respectively the equatorial radius and the widening of the barrel—Protein Engineering, 1990; Iraq and Syria have been ruled by small minorities (Sunni Arabs and Alawite officers respectively) — Internat. Affairs, 1991; The product is a joint venture by Rudy Rucker, a science fiction writer, and John Walker, who are employed respectively as 'mathenaut' and 'virtual programmer' by the computer-aided design specialist Autodesk Inc.—New Scientist, 1991.”